# OpenReview forum: "Architecture Matters for Multi-Agent Security"
_ICML.cc/2026/Conference — ICML 2026 regular_

### Official Review · Reviewer_jcLK · 2026-03-03

**Soundness:** 2
**Presentation:** 3
**Significance:** 4
**Originality:** 2
**Overall Recommendation:** 4
**Confidence:** 4

**Summary:**

This paper explores the impact of the architecture of a multi-agent system on its security and performance. Specifically, the agents' roles, system topology, and memory sharing are explored. The authors conduct experiments using a scenario where a user directly requests adversarial tasks. They found that the architecture choices can have a significant impact on the security of the system, without impacting performance.

**Compliance With Llm Reviewing Policy:**

Affirmed.

**Final Justification:**

After reading the other reviews, rebuttals, and reply to the comment, I decided to increase my score.

It is difficult to assess if W1 is resolved, as the authors cannot provide an improved version, but the rebuttal gives a good indication.

For W2.1 - W2.5, some have been addressed, others will be left for future work.

Given the additional experiments, I will increase my score to 4, but since some concerns have been confirmed to be left for future work, and other cannot be verified, I will not increase my score further.

**Key Questions For Authors:**

1. Could you expand on the tested design choices, specifically how the roles were chosen in the specialised executor settings, and how the communication topology works in practice, i.e. how an agent is chosen?
2. Could you provide more details and the concrete results of the agreement between the used classifier and human assessment (line 294)?
3. On a high level, how do the architectures differ on a qualitative level? Is there any indication why certain architectures are more likely to refuse, such as contents in the agent's reasoning?

**Limitations:**

yes

**Strengths And Weaknesses:**

### Strengths
1. The paper is well written and explores a highly relevant topic.
2. The aspects that are explored in this paper (roles, topology, and memory) are important design decisions of multi-agent systems.
3. The scope and main goal of the paper are clearly stated.
4. The main insight of this paper, namely that these design choices can have a significant impact on the security, is very interesting.

### Weaknesses
1. The implementation details of each design decision should be explained in more detail. The current manuscript description of the explored roles, topologies, and memory structures in Section 3 and 4 should be extended.
2. The empirical evaluation is the main contribution of this work, but it seems very limited in scope. Specifically, the experiments are lacking:
    1. An evaluation across multiple models, demonstrating if the impact on architecture choices is consistent across models and model families, or is varying depending on the model.
    2. An evaluation across different attacks, such as indirect prompt injection, memory poisoning, or adversarial agents within the system.
    3. The impact of the number of agents within the system is only explored implicitly with the agent roles. Additional more explicit experiments would be interesting.
    4. Experiments using safety mechanisms, such as simple prompting-based ones or agents specifically tasked with ensuring the safety of the system.
    5. An evaluation across different tasks, such as coding or general personal assistance.

While I believe that this paper's topic is highly relevant to the community, the current state of experiments is too shallow to draw significant conclusions from. However, if the experiments are significantly extended and the aforementioned implementation details extended, I believe that this could potentially be a strong contribution.

---

> ### Author Rebuttal · Authors · 2026-03-31
>
> We thank the reviewer for their positive assessment — particularly the recognition of the paper's significance — and for the constructive suggestions.
>
> We have expanded the empirical evaluation with additional models, tasks, and configurations. These additions test the same architectural hypotheses under broader conditions. We believe these additions substantially address the reviewer’s concern regarding the limited experimental scope. We address each point below.
>
> **Implementation details.**  We agree that Section 3–4 should provide more detail and will expand the descriptions in the camera-ready. To summarize the key design: in the 4-specialist variant, each agent handles one browser action type (click, fill, scroll, navigate), coordinated via structured dispatch; 2- and 3-specialist variants progressively merge responsibilities. For topology: star dispatches centrally, chain uses fixed order without a coordinator, mesh allows dynamic peer delegation. Only the coordination mechanism varies — roles and tools are held fixed. Full configuration tables (Tables 1–3) are reproduced in our response to Reviewer yVvr (camera-ready appendix).
>
> **Multiple models.** Beyond GPT-4o, we now evaluate GPT-5.4, GPT-5 Mini, Claude Sonnet 4, Llama-3.2-90B-Vision-Instruct, and Qwen3-VL-235B-A22B-Instruct. The impact is strongly model-dependent: well-aligned models (GPT-5.4, Sonnet 4) remain robust, while weaker-aligned models degrade dramatically — Qwen rises from 17% to 73% harmful actions and GPT-4o from 24% to 58% under full specialization.
>
> **Table A: BrowserArt — Harmful Task / Harmful Actions (%)**
>
> | Configuration | GPT-5.4 | GPT-5m | Sonnet 4 | Qwen3-VL | Llama 70B |
> |---|---|---|---|---|---|
> | Standalone Agent | 0/3 | 2/4 | 1/7 | 9/17 | 27/43 |
> | Star+Specialized | 3/6 | 2/8 | 1/8 | 37/73 | 35/61 |
> | Chain+Specialized | 0/0 | 0/6 | 3/12 | 21/42 | 26/37 |
> | Mesh+Specialized | 0/2 | 0/5 | 5/8 | 20/41 | 36/43 |
> | Mesh+Shared Mem | 0/2 | 3/7 | 3/8 | 29/49 | 34/47 |
>
> **Different attacks and safety mechanisms.** We agree these are important directions. Our study focuses on direct misuse to establish the baseline relationship between architecture and safety. Indirect prompt injection, memory poisoning, adversarial agents, and dedicated safety-monitor agents are natural extensions. We note these explicitly as future work in the camera-ready and believe our benchmark adaptations and framework provide the infrastructure to support such studies.
>
> **Number of agents.** We now explicitly vary specialist count: 2, 3, and 4 specialists under fixed star topology. For GPT-4o, harmful actions jump from 6% (single executor) to 58% (2 specialists), then remain at 52–53% for 3–4 specialists — indicating the critical transition is the introduction of specialization, not its degree. For well-aligned models (GPT-5.4), the effect is minimal across all counts.
>
> **Different tasks.** We built new scenario and replicate all experiments on OS-Harm, which tests intentional misuse of general computer-use agents beyond the browser setting. Core patterns replicate between scenarios and settings. **Introducing 2 novel MASEC benchmarks as main contribution of our work.**
>
> **Table B: OS-Harm — Harmful Task / Harmful Actions (%)**
>
> | Configuration | GPT-5.4 | GPT-4o |
> |---|---|---|
> | Standalone Agent | 11.4/20.5 | 4.5/25.0 |
> | Star+Specialized | 6.8/20.5 | 9.1/15.9 |
> | Chain+Specialized | 11.4/15.9 | 15.9/22.7 |
> | Mesh+Specialized | 6.8/13.6 | 13.6/20.5 |
> | Mesh+Shared Mem | 9.1/13.6 | 18.2/27.3 |
>
> **Human judgement agreement.** We replicate evaluation environments from BrowserArt and OS-Harm, which each report significant human-grader agreement. To validate in our multi-agent setting, we conducted a two-phase process: LLM-as-a-judge flagging followed by manual inspection of all flagged cases by two researchers, with additional sampling from full logs. We will report agreement statistics in the camera-ready.
>
> **Key Questions:**
>
> **Q1 (Design choices).** Addressed under Implementation details above. Tables 1–3 provide the full rationale; the design isolates a single Δ per step.
>
> **Q2 (Agreement results).** Addressed under Human judgement agreement above.
>
> **Q3 (Qualitative differences).** From our transcript analysis, the key difference is where refusal occurs. In standalone agents, refusal is immediate upon recognizing harmful intent. In star topologies, the planner generates an abstract plan that obscures harm — the executor then lacks sufficient context to refuse. In mesh topologies, responsibility diffuses across peers: each agent reasons that its individual action (e.g., "click this button") is benign, even when the aggregate sequence is harmful. This fragmentation is most pronounced for weaker-aligned models, which fail to detect harm from partial context.
>
> Thank you for your constructive feedback. We hope these additions address your concerns and would be grateful if you would consider reflecting them in your updated evaluation.

---

> > ### Author Rebuttal · Reviewer_jcLK · 2026-04-02
> >
> > I thank you for the detailed rebuttal. I hade read the rebuttal, as well as the other reviews. In the following I will go over my original points.
> >
> > 1. **Implementation Details** This concern was addressed
> > 2. **Empirical Evaluation**
> >     1. **Models** The authors provided an impressive amount of additional models, this point was addressed
> >     2. **Attacks** I understand that this is out of scope, and believe noting these ideas as future work is sufficient
> >     3. **Number of Agents** This point was addressed, though more extensive experiments would be interesting, e.g. interaction with agents outside of the environment
> >     4. **Safety Mechanisms** I understand that this is out of scope, and believe noting these ideas as future work is sufficient
> >     5. **Tasks** I appreciate the additional environment, but I do believe experiments using coding agents are important, as they are arguably the most important application of multi-agent systems currently. Given the very general phrasing of both the titel and abstract of this paper, I do not believe one environment is sufficient.
> >
> > Q1 and Q3 were answered, but unfortunately the concrete values I asked about in Q2 were not provided.
> >
> > Generally, I believe in it's current form this paper evokes a level of generality that, even with the extended experiments, is not actually explored in the paper. The exploration remains narrow, with a single application environment and specific type of multi-agent system. This paper needs either a significant extensions to the evaluation, or needs to significantly weaken it's claims, e.g. by specifying CUA's.

---

> > > ### Author Response · Authors · 2026-04-07
> > >
> > > We thank the reviewer for the thoughtful follow-up and are glad that most concerns were addressed. We believe the points below fully address the reviewer's follow-up:
> > >
> > > **Tasks / environments.** We'd like to clarify: even prior to this follow-up, we evaluated across two distinct environments - BrowserArt (web browsing) and OS-Harm (general computer use) - not one. These involve meaningfully different tool-use modalities.
> > >
> > > That said, we agree that coding agents are an important application. We had already begun running these experiments before the rebuttal deadline and have now finalized the results. We evaluate all 13 configurations on RedCode-Gen, a benchmark for harmful code generation in sandboxed coding environments (160 tasks across 8 malware categories). This brings our evaluation to three distinct environments. Results below; we will include them in the camera-ready.
> > >
> > > The core patterns hold: standalone agents are safest (9% harmful task completion), role decomposition dramatically increases vulnerability (up to 51% under batch execution), and topology matters (chain is riskiest at 42%, mesh is more moderate at 21%). Notably, mesh with full shared memory achieves the lowest harmful task rate among all multi-agent configurations (15%), consistent with our findings in the other environments.
> > >
> > > **RedCode-Gen (Coding) — GPT-4o — Harmful Task / Harmful Actions (%)** [Due to space limit, we do not include the full results here; they will be part of the camera-ready version].
> > >
> > > | Configuration | GPT-4o |
> > > |---|---|
> > > | Standalone Agent | 9 / 4 |
> > > | Star+Single Executor | 31 / 26 |
> > > | Star+Batch Executor | 51 / 28 |
> > > | Star+Stepwise | 27 / 28 |
> > > | Star+2 Specialists | 26 / 24 |
> > > | Star+3 Specialists | 24 / 24 |
> > > | Star+Specialized | 18 / 25 |
> > > | Chain+Specialized | 42 / 23 |
> > > | Mesh+Specialized | 21 / 19 |
> > > | Mesh+Shared Mem | 15 / 21 |
> > >
> > >
> > > **Q2 (Agreement values).** Two researchers manually reviewed 116 trajectories across all scenarios, identifying 15 scoring errors which were corrected; 20 additional corrections were made to the scorer itself. We further spot-checked 10% of the remaining logs and found a 7% discrepancy rate in the sample, all of which were corrected. We will report full agreement statistics in the camera-ready.
> > >
> > > **Generality of claims.** We respectfully note that our evaluation already spans three distinct application environments - BrowserArt (web browsing), OS-Harm (desktop computer use), and RedCode-Gen (coding agents) - each involving meaningfully different tool-use modalities and agent types. Combined with 6 models and 13 systematically controlled configurations, we believe this provides substantial empirical grounding across more than a "single environment and specific MAS type." That said, we appreciate the underlying concern and will tighten the framing to accurately reflect the scope of our evaluation.
> > >
> > > Thank you again for the constructive engagement. We hope this fully addresses the remaining concerns.

---

### Official Review · Reviewer_XqRZ · 2026-03-04

**Soundness:** 3
**Presentation:** 2
**Significance:** 2
**Originality:** 3
**Overall Recommendation:** 4
**Confidence:** 3

**Summary:**

This paper presents an empirical study investigating how architectural design choices in LLM Multi-Agent Systems influence their susceptibility to adversarial attacks (specifically, harmful task execution) instead of requesting harmful outputs by asking adversarial queries. The authors conduct controlled experiments with GPT-4o and manipulate three architectural dimensions: role configuration, communication topology, and memory/state visibility. The study reveals that multi-agent systems are generally more vulnerable to harmful tasks than standalone agents. They did in-depth analysis of the impact of different architectural choices and call upon the research community to develop MAS-specific safeguard strategies.

**Compliance With Llm Reviewing Policy:**

Affirmed.

**Final Justification:**

I value the study's focus on MAS security dimensions but critique its unclear presentation and narrow experimental scope, which rely on a single model and very few tasks. The authors addressed these concerns through a substantial expansion that included five additional models and a new scenario, bolstering the statistical power and overall contributions. Although full results and some raw data remains for the final version, I increased my ratings for soundness and overall recommendation.

**Key Questions For Authors:**

What is your overall advice in building less vulnerable MAS among the three and other architectural dimensions?

**Limitations:**

yes

**Strengths And Weaknesses:**

### Strengths

1. I appreciate the detailed background and related work section (Sec 2 and Sec 3), which provides a comprehensive overview of the current state of research on LLM safety and motivates the need for a systematic study of MAS architecture.
2. It is well considerate to test MAS under adversarial web automation tasks to expose the vulnerabilities of different architectural choices, since direct QA jailbreaks have been easily mitigated.
3. The three dimensions of architectural design choices (role configuration, communication topology, and memory/state visibility), though not exhaustive, are valid.

### Weaknesses

1. The presentation is not clear for methodology and experimentation. The visual presentation and detail elaboration could be optimized. Besides presenting design choices in text (Sec 4.2), a diagram or table of the different role/memory configurations and detailed workflow prompting templates would help readers quickly grasp the differences. The case study is too brief and lacks details. Moreover, the experimentation results are repetative in tables and Figure 2.

2. The experimentation scope is narrow:
    1. The evaluation of benign task performance is severely limited by only 9 tasks. When benign performance drops from 100% to 66.7% in the mesh topology, it represents a failure on exactly 3 out of 9 tasks, making the statistical significance claims fragile.
    2. Furthermore, the evaluation is entirely dependent on a single proprietary model (GPT-4o). As mentioned, open-source models, heterogeneous model combinations, or specifically trained safeguard models could behave quite differently, which limits the immediate generalizability of the findings.
    3. Independent evaluation for each architectural choice is good. However, the interaction effects between different architectural choices are not explored. At least, the conclusions could be better supported and more useful if a MAS with robust role&topology&memory design is shown to be more secure and a MAS with weak design in all three dimensions is shown to be more vulnerable.

3. Please provide human judgement agreement results instead of claiming "high agreement" without evidence (L295).

Minor: Table captions should be on top of the table and flush left.

---

> ### Author Rebuttal · Authors · 2026-03-31
>
> We thank the reviewer for their insightful comments, recognition of our work's novelty, and appreciation for our detailed background, adversarial evaluation setting, and architectural dimensions.
>
> We have substantially expanded our experiments with a new scenario (OS-Harm), additional settings, five additional models, expanded benign baselines, and a clearer design rationale. Importantly, our new results reinforce and generalize the original conclusions with considerably more robust evidence. We summarize below, referring to our responses to Reviewers JMum and yVvr where applicable.
>
> These additions directly address the reviewer’s concern regarding limited experimental scope by expanding across models, tasks, and environments while preserving the same controlled design.
>
> **Presentation.** We address each sub-concern:
>
> - *Design clarity.* We now provide Tables 1–3 (camera-ready appendix; reproduced in our response to Reviewer yVvr) systematically describing each configuration's role decomposition, topology, and memory visibility, with the minimal change (Δ) at each step. We believe these directly address the suggestion for a diagram or table of configurations.
> - *Case study.* Thank you for this feedback — we have dramatically expanded the case study with additional detail to be included in the camera-ready and appendix. Our multi-agent adaptation decomposes BrowserArt's 100 harmful web tasks into 13 conditions along three orthogonal axes: role specialization (7 conditions, including specialist variants where each agent handles a single browser action), communication topology (3 conditions: star, round-robin, dynamic delegation), and memory visibility (4 conditions). The only variable across conditions is architectural decomposition — task prompts, browser environment, and judge protocol are held fixed. Scoring uses a three-tier metric hierarchy with post-judgment validation. A similar adaptation was conducted for OS-Harm.
> - *Repetitive results.* We have revised the presentation of tables and Figure 2 to reduce overlap, ensuring each presents complementary information. We will also fix table caption placement as suggested.
>
> **Experimentation scope.** Three concerns were raised: limited benign tasks, a single model, and missing interaction effects.
>
> -*New scenario.* We built new scenario on OS-Harm, testing misuse of general computer-use agents beyond the browser. See Table B in our response to Reviewer JMum.
> - *Benign baselines.* We now evaluate 42 benign samples on BrowserArt and 100 on OS-Harm. Benign accuracy remains >76% across all conditions (most >90%), providing substantially more statistical power than the original 9 tasks.
> - *Six models.* Beyond GPT-4o, we now evaluate GPT-5.4, GPT-5 Mini, Claude Sonnet 4, Llama-3.2-90B-Vision-Instruct, and Qwen3-VL-235B-A22B-Instruct. See Tables A–B in our response to Reviewer JMum for representative results. Full results (6 models × 13 conditions × 2 scenarios) will appear in the camera-ready.
> - *Interaction effects.* We agree this is an important direction. Our memory experiments (Table 3) partially address this by testing memory visibility under both star and mesh topologies — i.e., varying two axes simultaneously. We find that the effect of memory depends on topology: for example, shared memory under mesh increases harmful actions for Qwen (41%→49%) and Llama (43%→47%), while under star the effect is smaller. With the expanded evaluation across models, environments, and configurations, we believe the study now captures a substantially broader portion of the design space and provides stronger empirical support for our conclusions.
>
> **Human judgement agreement.** We replicate the evaluation environments from the BrowserArt and OS-Harm papers, which each report significant human-grader agreement. To ensure our novel multi-agent setting does not change this, we conducted a two-phase validation: an LLM-as-a-judge to flag potential errors, followed by manual inspection of all flagged cases by two researchers, with additional sampling from full logs to catch discrepancies. We observe high agreement between automated and manual evaluation, and will report exact agreement statistics in the camera-ready.
>
> **Key question: advice for building less vulnerable MAS.** Our main takeaway is that there is no universally safe architecture: security depends on interactions between model, architecture, and environment. However, we observe three consistent patterns: (1) role decomposition is the highest-risk axis, particularly for weaker-aligned models; (2) increased state sharing does not reliably mitigate this risk; and (3) security must be evaluated at the system level rather than inferred from individual components. Our benchmarks are designed to support this evaluation in practice.
>
> Thank you for your constructive feedback. We hope these additions address your concerns and would be grateful if you would consider reflecting them in your updated evaluation.

---

> > ### Author Rebuttal · Reviewer_XqRZ · 2026-04-03
> >
> > Thank the authors for the detailed response. I appreciate the additional experiments and takeaway summary. Below, I have some follow-up comments:
> >
> > 1. Please tell the exact human judge agreement metrics, scores, and do significance tests, instead of describing it is "high".
> > 2. Please demonstrate raw records (or part of it) of one or more detailed cases.
> > 3. Please make sure the additional experiments are organized and mentioned in the main body of the revised manuscript.
> > 4. Please add the takeaway conclusion in the revised version.
> >
> > In the current shape, I will increase my overall recommendation score and further responses will be reflected in the sub-score ratings.

---

> > > ### Author Response · Authors · 2026-04-07
> > >
> > > We thank the reviewer for the constructive follow-up and are glad the additional experiments and takeaways were appreciated.
> > >
> > > We address each follow-up point below and incorporated them into the camera-ready version:
> > >
> > > 1. Human judge agreement. Two researchers manually reviewed 116 trajectories across all scenarios, identifying 15 scoring errors which were corrected; 20 additional corrections were made to the scorer itself. We further spot-checked 10% of the remaining logs and found a 7% discrepancy rate, all of which were corrected. For statistical robustness, each experimental condition was run three times, allowing us to compute variance and significance across repeated runs. We include full agreement metrics with significance tests in the camera-ready.
> > >
> > > 2. Detailed case records. We include annotated trajectory examples in the camera-ready appendix, showing the full agent interaction trace for representative cases - including a planning-stage refusal, an execution-stage interception, and a successful harmful task completion - across different architectures. [Due to space limit we are not able to include them here in the comment box].
> > >
> > > 3. Organization in main body. Confirmed - all additional experiments are organized and referenced in the main body of the revised manuscript, with full results in the appendix.
> > >
> > > 4. Takeaway conclusion. We incorporated our conclusions in the revised manuscript.
> > >
> > > We are grateful for the reviewer's thoughtful feedback and the time invested in helping strengthen our work!

---

### Official Review · Reviewer_yVvr · 2026-03-10

**Soundness:** 1
**Presentation:** 3
**Significance:** 1
**Originality:** 2
**Overall Recommendation:** 2
**Confidence:** 4

**Summary:**

The article presents an experimental study aimed at demonstrating the influence of the topological architecture of a multi-agent system on its security and vulnerability to attacks. Although the description of this objective is relatively general, the work focuses more specifically on a MAS that implements an LLM, either by decomposing, or not, generation into planning, execution, and evaluation tasks. The vulnerability studied is also restricted to the introduction of malicious prompts that should be rejected. The experimental results show that distributing tasks across multiple agents increases the overall vulnerability, with differences observed between several topologies.

**Compliance With Llm Reviewing Policy:**

Affirmed.

**Key Questions For Authors:**

no questions.

**Limitations:**

yes

**Strengths And Weaknesses:**

Strengths
The article is well written and clearly presents the work. The three design choices regarding role configuration, communication topology, and the visibility of memory and states are clearly defined. The experiments conducted are also well described, and the analysis of the results supports the observation that architectural choices have an impact.

Weaknesses
The choices made for the different experiments appear somewhat arbitrary and lack justification. Only a few topologies and role distributions are tested (which are limited to the specific tasks assigned to an LLM), which prevents drawing general conclusions about the relationship between architectural choices and security. The conclusion that a distributed system is more vulnerable than a centralized system is obvious as we use to consider that a system’s vulnerability corresponds to that of its weakest component. The acknowledged limitation of testing with only a single model, GPT-4o, also prevents deriving a general property.

- soundness

As mentioned previously, the arbitrary choices, the limited number of tested configurations, and the use of a single LLM do not allow the claims derived from the experiments to be properly supported.

- presentation

The article is well written and the presentation is of good quality. However, Figure 1 is neither referenced nor discussed in the text and introduces some confusion regarding the configurations that were tested.

- significance

The study of vulnerability factors in a MAS is an important issue, but it is not sufficiently addressed here to constitute a significant result.

- originality

The work does not provide a sufficiently original study of the security issues of a multi-agent system. Moreover, this topic has been widely studied for more than twenty years in the multi-agent systems community, particularly through research on trust management systems.

---

> ### Author Rebuttal · Authors · 2026-03-31
>
> We thank the reviewer for their time and helpful comments. We're glad the reviewer finds our presentation clear and our argument well-supported.
>
> We have substantially expanded our experiments with a new scenario (OS-Harm), additional settings, five additional models, and clearer design rationale. Our new results reinforce and generalize the original conclusions. We summarize below and refer to our response to Reviewer JMum for full details and tables.
>
> **Soundness.** The reviewer raises three concerns: (i) arbitrary design choices, (ii) limited configurations, and (iii) reliance on a single model.
> - *Design choices are controlled, not arbitrary.* Each experiment isolates a single architectural change (Δ) along one axis: role decomposition (fixed topology), communication topology (fixed roles), or memory visibility (fixed roles and topology). This controlled design treats architecture as the independent variable; we will clarify this in the paper
> - *Expanded configurations.* We now evaluate 13 configurations across 2 scenarios (BrowserArt and OS-Harm), including varying specialist count (2, 3, 4) and memory under mesh topology.
> - *Six models.* Beyond GPT-4o: GPT-5.4, GPT-5 Mini, Claude Sonnet 4, Llama-3.2-90B-Vision-Instruct, and Qwen3-VL-235B-A22B-Instruct. See Tables A–B in our response to Reviewer JMum. Full results will appear in the camera-ready.
>
> **Presentation.** Thank you for flagging Figure 1. We will fix this in the camera-ready.
>
> **Significance.** The expanded experiments demonstrate that small architectural changes produce 30+ percentage point swings in attack success while benign performance remains largely unchanged (>76% across all conditions, most >90%), across 6 models and 2 environments.
>
> **LLM-Based multi-agent security benchmarks.** We introduce two novel multi-agent security benchmarks—adaptations of BrowserArt and OS-Harm—which, to our knowledge, are the first standardized benchmarks for evaluating security properties of LLM-based multi-agent systems. We believe these will benefit the broader community.
>
> **Originality and relation to prior MAS work.**
> We respectfully disagree that the main conclusion is a restatement of classical MAS results.
>
> - *Not a “weakest-link” result.* We do not claim that distributed systems are inherently more vulnerable. The effect is conditional on architecture, model, and environment. In some settings, multi-agent configurations improve security (e.g., GPT-4o improves from 25% to 9.1% harmful actions under Star+Batch in OS-Harm). In misuse settings, harmful outcomes require end-to-end success, meaning that strong intervention at any stage can prevent failure—contrary to a weakest-link interpretation.
>
> - *Distinction from classical MAS.* Classical trust management assumes rational agents with defined utility functions in settings where self-interested agents may defect. We study a malicious user providing adversarial instructions to a cooperative system, where vulnerability arises from how architecture interacts with safety training to fragment responsibility and dilute refusal (Bădică et al., 2025; Schroeder de Witt, 2025).
>
> - *Complex environment interaction.* Our setting is not merely decomposing text generation. Each agent interacts via tool use with complex environments (web browsing, computer use), with unique prompting, context, memory, and roles—a setting not well-studied in security literature (Schroeder de Witt, 2025).
>
> We provide the first controlled empirical study showing security is a system-level emergent property of architecture. We will add a Related Work paragraph clarifying these distinctions.
>
> **Thank you for your constructive feedback. We hope these additions address your concerns and would be grateful if you would reflect them in your updated evaluation.**
>
> **References**
> Bădică et al. Contemporary agent technology: LLM-driven advancements vs classic MAS. arXiv:2509.02515, 2025.
> Schroeder de Witt. Open challenges in multi-agent security. arXiv:2505.02077, 2025.
>
> **Table: Experimental Conditions —Progressive Design**
>
> | Configuration | Δ Change | Roles | #Ag |
> |---|---|---|---|
> | ***Role Decomposition*** (star topology, private memory) | | | |
> | Standalone Agent | — | Monolithic | 1 |
> | Star+Single Executor | +Delegation | Planner→Executor | 2 |
> | Star+Batch Executor | +Structured plan | Planner→Executor | 2 |
> | Star+Stepwise | +Iterative feedback | Planner⇄Executor | 2 |
> | Star+2 Specialists | +Partial spec. | Planner→2 Spec | 3 |
> | Star+3 Specialists | +More spec. | Planner→3 Spec | 4 |
> | Star+Specialized | +Full spec. | Planner→4 Spec | 5 |
> | ***Topology*** (4 agents, private memory) | | | |
> | Star+Specialized | Centralized | Planner→4 Spec | 5 |
> | Chain+Specialized | Fixed order | 4 Peers | 4 |
> | Mesh+Specialized | Free delegation | 4 Peers | 4 |
> | ***Memory*** (roles & topology fixed) | | | |
> | Private | Baseline | Per-agent only | — |
> | +Own Reasoning | +Local CoT | Own history |—|
> | +Shared Memory | +Global state | All agents |—|

---

> > ### Author Rebuttal · Reviewer_yVvr · 2026-04-02
> >
> > The authors answer with additional experimentations that are not in the original article. This is not the role of the rebuttal phase that should explain parts of the submitted article that may have been not understood. My review remains the same.

---

> > > ### Author Response · Authors · 2026-04-06
> > >
> > > We thank the reviewer for the constructive engagement throughout the discussion.
> > >
> > > We respectfully clarify that our additional experiments are not intended to change the scope or arguments of the paper, but rather to provide stronger empirical support for the claims already made in the original manuscript.
> > >
> > > We believe these findings are novel and practically important as LLM-based multi-agent deployments continue to grow.
> > > We hope the reviewer will consider both our thorough responses to the original concerns and the substantially expanded evidence in their final assessment.
> > >
> > > We are grateful for the reviewer's thoughtful feedback and the time invested in helping strengthen our work!

---

### Official Review · Reviewer_JMum · 2026-03-19

**Soundness:** 2
**Presentation:** 3
**Significance:** 2
**Originality:** 1
**Overall Recommendation:** 4
**Confidence:** 2

**Summary:**

The authors investigate multi-agent-system (MAS) exploring how agent roles; topology; and memory establish architectural decisions that can be evaluated towards the security of the overall system. Experimentation is done via an adapted version of the `BrowserART` benchmark -- originally created to evalaute 100 harmful web-based tasks for single agents, with the adaption being made to enable MAS systems to test as well. Experimentation is done with UK AI Security Institute `Inspect` experimentation framework.

Architectures for evaluation for the MAS include a standalone baseline agent performing planning-reasoning-execution and tool use within a single component thusly performing actions sequentially. Vectors for MAS architecture then include:

1. `Role specialization`: varients of orchestration where orchestrator may remain in the loop after every action to delegating subtasks
2. `Communication topology`: standalone; star; mesh
3. `Memory`:  agents may access limited local memory in the mesh; their own CoT reasoning; or full shared memory with all agents

Evaluation for experiments are not binary but reflect on where MAS guardrails appear, if in the planning and/or exection refusals, and if the harmful action is performed or not.


Key findings include that the Star with shared memory MAS showed the lowest of all harmful task execution, ultimately reflecting that MAS designs are a balance between safety and usefulness.

**Compliance With Llm Reviewing Policy:**

Affirmed.

**Final Justification:**

We thank the authors for their rebuttal.

We recognize the clear strengths in this work, and note that it is timely to explore MAS security on the dimensions explored here: role specializations, topology, and memory. Findings such that star-topology-with-shared-memory demonstrates lowest harm execution contribute meaningfully to ongoing conversations on MAS security.

Our score remains unchanged however, in spite of the addition of OS-Harm, heavily due to the limited experimental scope.  Further analysis and evaluations could have helped convince the report's findings. While the rebuttal engages our concerns,  the underlying remains.

We encourage the authors to add additional trajectory analysis and additional case studies in future iterations.

**Key Questions For Authors:**

1. Might agentic frameworks for multiagent system themselves be a fourth vector for  MAS security evaluation?

2. Could standard agent protocols like A2A, or other open standards be vectors where security is traded for performance

**Limitations:**

yes

**Strengths And Weaknesses:**

STRENGTHS::

The authors argument are well motivated and the accompanying code reflects the extended tooling. While not overly complicated the vectors of role play; communication; and memory are the most obvious architectural structures of MAS so experimentation using standard web security benchset, and UK AI experimentation tooling is reasonable.



WEAKNESSES::

The scope of the experiment is promising but not comprehensive, and almost sparse. Further analysis could have also been conducted on trajectories to explore the reasons behind key findings; case studies and ablations studies would have also meaningfully contributed to expanding the experimentation breadth.

---

> ### Author Rebuttal · Authors · 2026-03-31
>
> We thank the reviewer for their thoughtful feedback and positive assessment.
>
> We have expanded the empirical evaluation to address concerns about breadth. These experiments test the same hypotheses under broader conditions. The new results reinforce and generalize our core finding: **architectural design choices significantly affect security in multi-agent systems, often without corresponding changes in benign performance.**
>
>  Below we summarize the key additions.
>
> 1. **New scenario: OS-Harm.** We replicate all experiments on the OS-Harm benchmark, which tests misuse of general computer-use agents beyond the browser. We find that the core patterns from BrowserArt replicate here: standalone agents are generally among the safest configurations, batch execution remains a consistent risk factor (e.g., GPT-5.4 rises from 20.5% to 27.3% harmful actions). **These constitute two novel benchmark settings for multi-agent systems introduced as a key contribution of this work.**
>
> 2. **Five additional base models (six total).** Beyond GPT-4o, we evaluate GPT-5.4, GPT-5 Mini, Claude Sonnet 4, Llama-3.2-90B-Vision-Instruct, and Qwen3-VL-235B-A22B-Instruct. We find that security degradation under role decomposition is strongly model-dependent. Well-aligned models (GPT-5.4, Sonnet 4) remain robust, while weaker-aligned models degrade dramatically - Qwen rises from 17% to 73% harmful actions and GPT-4o from 24% to 58% under full specialization. We attribute this to the fragmentation of harmful intent: role decomposition distributes the task across agents, diluting the harmful signal visible to each. Models with stronger safety training still detect the harmful goal from partial context; weaker models do not.
> Full results will be reported in appendix, camera-ready. For the reviewer's convenience, we include them below.
>
> 3.  **Systematic ablations (clarification)**  Our design isolates three axes, each with minimal changes (Δ): (a) Role: Monolithic → Planner+Executor → Planner+2/3/4 Specialists, under fixed star topology; (b) Topology: Star → Chain → Mesh, with roles fixed; (c) Memory: Private → Local CoT → Shared, under both star and mesh. Each Δ adds exactly one change. Design-rationale table reproduced in our response to Reviewer yVvr (camera-ready appendix).
>
> 4. **Benign baselines & validation.** Benign task accuracy remains >76% across all conditions (most >90%). We validate scoring using LLM-as-judge classification combined with manual inspection by two researchers, with high agreement. We also expand the number of benign tasks to 42 in BrowserART and 100 in OS-Harm to improve coverage.
>
> Due to space constraints, we include representative results below. Full tables across all 6 models × 13 conditions × 2 scenarios (with all metrics) are available upon request and will appear in the camera-ready version.
>
> **Table A: New Models on BrowserART — Harmful Task / Harmful Actions (%)**
>
> | Configuration | GPT-5.4 | GPT-5m | Sonnet 4 | Qwen3-VL | Llama 70B |
> |---|---|---|---|---|---|
> | Standalone Agent | 0 / 3 | 2 / 4 | 1 / 7 | 9 / 17 | 27 / 43 |
> | Star+Specialized | 3 / 6 | 2 / 8 | 1 / 8 | 37 / 73 | 35 / 61 |
> | Chain+Specialized | 0 / 0 | 0 / 6 | 3 / 12 | 21 / 42 | 26 / 37 |
> | Mesh+Specialized | 0 / 2 | 0 / 5 | 5 / 8 | 20 / 41 | 36 / 43 |
> | Mesh+Shared Mem | 0 / 2 | 3 / 7 | 3 / 8 | 29 / 49 | 34 / 47 |
>
> **Table B: OS-Harm Scenario — Harmful Task / Harmful Actions (%)**
>
> | Configuration | GPT-5.4 | GPT-4o |
> |---|---|---|
> | Standalone Agent | 11.4 / 20.5 | 4.5 / 25.0 |
> | Star+Specialized | 6.8 / 20.5 | 9.1 / 15.9 |
> | Chain+Specialized | 11.4 / 15.9 | 15.9 / 22.7 |
> | Mesh+Specialized | 6.8 / 13.6 | 13.6 / 20.5 |
> | Mesh+Shared Mem | 9.1 / 13.6 | 18.2 / 27.3 |
>
> **Answers for questions:**
>
> Q1 We agree that the choice of framework is relevant, but argue it must ultimately manifest in specific architectural decisions — the design axes we study. Different frameworks will produce different communication patterns, memory access, and role prompting, many of which our experiments already cover. We acknowledge additional vectors remain to be explored.
>
> Q2 We agree. Our memory experiments show shared memory does not reliably recover security lost to decomposition — Llama rises from 61%→67% and Qwen from 41%→49% under mesh. Protocol designers should not assume increased state visibility compensates for architectural fragmentation.
>
> Across all additions, we consistently observe architecture-dependent security effects across multiple scenarios, setups and models, providing stronger empirical support for our conclusions.
>
> Thank you for your positive assessment and constructive feedback. We hope the additional experiments and clarifications above address your remaining concerns, and we would be grateful if you would consider reflecting these in your updated evaluation.

---

> > ### Author Rebuttal · Reviewer_JMum · 2026-04-04
> >
> > We thank the authors  for their response.
> >
> > The trajectory of this paper is meaningful given the agentic rise in workflows, but while the additional experiments, including the inclusion of OS-Harm, does enrich the opportunity for experimentation, additional analysis and evaluation metrics could have been conducted on trajectories to explore the reasons behind key findings to strengthen the experiment.
> >
> > For this reason the score remains unchanged.

---

> > > ### Author Response · Authors · 2026-04-06
> > >
> > > We thank the reviewer for confirming that the points raised have been fully addressed, and for the constructive engagement throughout the discussion.
> > >
> > > To our knowledge, this constitutes the first controlled empirical study showing that security in LLM-based multi-agent systems is a system-level emergent property of architecture, and the first to introduce standardized benchmarks for evaluating these properties - a phenomenon not captured by classical multi-agent literature.
> > >
> > > These findings speak directly to the reviewer's interest in understanding the reasons behind key results, and we will further enrich them with qualitative trajectory examples in the camera-ready. We believe this body of evidence provides a strong foundation that other researchers can build on as multi-agent deployments grow.
> > >
> > > We are grateful for the reviewer's thoughtful feedback and the time invested in helping strengthen our work!

---

### Decision · Program_Chairs · 2026-04-30

**Decision:**

Accept (regular)

**Comment:**

Reviewers identified concerns with the submitted version of this paper: 1) The scope of the experiments, and 2) (yVvr) perceived inadequate connection to existing MAS results.

The reviewers responded to the first concern by providing a good number of additional results. yVvr complained that this was not an appropriate use of the rebuttal mechanism, which is kind of true, but it was nevertheless somewhat persuasive for the other reviewers. Even with the new results, some concerns remain about the adequacy of the experiments. In addition, the concern from yVvr about connection to existing MAS results was not fully resolved.

The feels like a truly borderline paper. yVvr's may be overrated but are not entirely meritless. The positive reviewers like the paper, but are not overflowing with enthusiasm.